# Species of the Western Palaearctic Genus *Tetralonia* Spinola, 1838 (Hymenoptera, Apidae) with Atypical Pollen Hosts, with a Key to the *pollinosa*-Group, Description of New Species, and Neotype Designation for *Apis malvae* Rossi, 1790 †

Achik Dorchin [1,2,*] and Denis Michez [1]

1   Laboratory of Zoology, Research Institute for Biosciences, University of Mons, Place du Parc 20, Mons 7000, Belgium; denis.michez@umons.ac.be
2   Department of Biology—Invertebrates, Royal Museum for Central Africa, Leuvensesteenweg 13, Tervuren 3080, Belgium
*   Correspondence: achik.dorchin@umons.ac.be
†   urn:lsid:zoobank.org:pub:0670F038-B376-4F11-BC21-BC27A0967989,
    urn:lsid:zoobank.org:act:B6F44D70-8285-4F66-A74E-0DEF9498E229,
    urn:lsid:zoobank.org:act:DC0EEBE8-A59F-4DB0-A0E1-9C8B63989FCA,
    urn:lsid:zoobank.org:act:CE87467C-B528-4501-ABF7-4FA54C1E69B0.

**Abstract:** The long-horn bee genus *Tetralonia* consists of 35 Western Palaearctic species that are associated mostly with the family Asteraceae as host plants. A minority of the species are, however, exclusively associated with other host plants that have particularly large pollen grains, such as those in the plant families Caprifoliaceae, Malvaceae, and Onagraceae. This work presents a taxonomic account and morphological description of the assemblages of *Tetralonia* species with atypical (non-Asteraceae) host plants. It includes a key to the *pollinosa*-group, which contains most of the species, a description of three regionally restricted new species, namely *T. eoacinctella* Dorchin **sp. nov.**, *T. epilobii* Dorchin **sp. nov.**, and *T. stellipilis* Dorchin **sp. nov.**, a lectotype designation for *Eucera cinctella* Saunders, 1908 [=*Tetralonia cinctella* (Saunders, 1908)], and a neotype designation for *Apis malvae* Rossi, 1790 [=*Tetralonia malvae* (Rossi, 1790)]. In addition, the name *Eucera macroglossa* Illiger, 1806 is confirmed as a synonym of *Apis malvae* Rossi, 1790; *Tetralonia macroglossa* ssp. *xanthopyga* Alfken, 1936 is officially placed in synonymy with *Apis malvae* Rossi, 1790; and *Macrocera confusa* Pérez, 1902 is listed as a doubtful synonym of *Tetralonia scabiosae* Mocsàry, 1879 (**syn. nov.**).

**Keywords:** taxonomy; systematics; classification; Palaearctic; Mediterranean; pollinators; solitary bees

## 1. Introduction

*Tetralonia* Spinola, 1838 [1] is a genus in the long-horn bee tribe Eucerini (Apidae), best known for the unusually long antennae of the males. *Tetralonia*, in its current delimitation, is restricted to the Eastern Hemisphere of the world, and typically includes warm climate species and species that are active in the summer in temperate climates [2,3]. The classification history of the genus has been convoluted, and much confusion persisted over the delimitation of *Tetralonia* with respect to the morphologically reminiscent, species-rich genus, *Eucera*, Scopoli, 1770 [4]. These two taxa were repeatedly synonymised [2], and *Tetralonia* was most recently reestablished as valid based on recent phylogenetic evidence [3,5]. Additional names that were previously associated with the genus are the synonym *Tetraloniella* Ashmead, 1899 [6] and the taxa *Xenoglossodes* Ashmead, 1899 [6] and *Synhalonia* Patton, 1879 [7], with the latter now being assigned to the genus *Eucera* [2,3]. Particularly in the Palaearctic region, where both *Tetralonia* and *Eucera* occur in sympatry,

the diagnosis of these taxa remains difficult, a condition which has probably hampered comprehensive taxonomic and systematic treatment.

About 35 of the 101 described species in the genus *Tetralonia* are known from the Western Palaearctic region [8]. They are typically oligolectic species, proven or assumed to specialize on the collection of pollen from a single botanical family, mainly the family Asteraceae [9]. The females of these species exhibit various morphological adaptations, like strongly branched scopal hairs that help to efficiently bind and transport the usually small and spiny pollen grains of the Asteraceae (while the males do not collect pollen and share other morphological characteristics). A minority of the species are associated with much larger pollen grains, such as those in the plant families Caprifoliaceae, Malvaceae, and Onagraceae [9]. Species with atypical (non-Asteraceae) floral hosts often converge on certain morphologies, but they do not form a single clade, rather they appear as several different lineages among the more ordinary Asteraceae specialists in the phylogenetic tree [9]. A comprehensive phylogenetic analysis of the genus *Tetralonia* is still lacking, and the taxonomy of the group is complicated, with many species being poorly characterized and some old names that remain little known [5]. As a first step toward resolving the taxonomy of the group, this paper provides diagnoses for the identification of the Western Palaearctic species with atypical (non-Asteraceae) host plants. A taxonomic account of the species is presented; diagnostic characteristics are provided together with a key to the *pollinosa*-group, a group of Caprifoliaceae specialists that comprises the majority of the species; three regionally restricted species are newly described; and a neotype is designated for *Apis malvae* Rossi, 1790 [10] to fix its status as the oldest name of the type species of the genus *Tetralonia*.

## 2. Materials and Methods

The terminology used for describing the morphology of the species follows that of Michener [11], including the abbreviations for metasomal tergites and sternites, T1–7 and S1–8, respectively. Dorchin et al.'s [2] terminology is used for the structurally complex genitalia and S6–8 of the male. These particularly include the converging carinae of S6, comprising a posterolateral oblique carina and sometimes an anterior carina or ridge; the lateral and posteromedial processes of S7; and the apical lobes of S8. For examination, male sternites and genitalia were removed and cleared in NaOH 1N, then washed in 100% ethanol and mounted onto a white mounting board using fine forceps. Images were taken with the Keyens VHX-970F imaging system at 50–200× magnification. Habitus images of the new species were taken with an Olympus E-M1 Mark II camera equipped with the Olympus Zuiko 60 mm F:2.8 macro lens (and with an additional Marumi lens + 5 dioptries, 55 mm). All other images were taken with an additional Mitutoyo plan achromatic LWD 5× (or 10×) objective. The images were stacked with the software Helicon Focus v8.2.7, edited using GIMP v2.8.18, and plates were prepared using Inkscape v1.3.

The species descriptions provide detailed diagnoses of the most conspicuous morphological features that are useful for the identification of each of the species. Body and forewing lengths were measured using a metal ruler under a magnification of 6.4–16× to a precision of 0.5 mm based on 2–5 female and male specimens from different localities. Body length represents the distance from the apex of the clypeus to that of the pygidial plate of specimens in which the head was perpendicular to the main axis of the body. Forewing length was measured as the longest distance from the margin of the tegulae to the apex of the wing.

The species are presented by group in alphabetic order, and a list of synonyms is provided in chronological order followed by comments (or references are cited) on each of the species names examined. Descriptions of new species are presented separately.

The following abbreviations are used for depository institutes of type material.

ADCM—Achik Dorchin research collection, University of Mons, Belgium;
BMNH—The Natural History Museum, London, UK;
ISEAP—Polish Academy of Sciences, Institute of Systematics and Evolution of Animals, Krakow, Poland;
MNB—Museum für Naturkunde—Leibniz Institute for Evolution and Biodiversity Science, Berlin, Germany;
MNHN—Muséum national d'Histoire naturelle, Paris, France;
MTM—Hungarian Natural History Museum, Budapest, Hungary;
NHMD—Natural History museum of Denmark, University of Copenhagen, Denmark;
OLML—Oberösterreichisches Landesmuseum, Linz, Austria;
OUM—Oxford University Museum of Natural History, UK;
SMNH—The Steinhardt Museum of Natural History, Tel Aviv University, Israel;
ZINSP—Russian Academy of Sciences, Zoological Institute, St. Petersburg, Russia.

## 3. Results

### *3.1. pollinosa-Group*

#### 3.1.1. Diagnosis

The group includes the species *T. pollinosa* (Lepeletier, 1841) [12], *T. strigata* (Lepeletier, 1841) [12], *T. scabiosae* Mocsàry, 1879 [13], and *T. cinctella* (Saunders, 1908) [14] and two species are newly described in this work, *T. stellipilis* **sp. nov.** and *T. eoacinctella* **sp. nov**. Members of the group are associated with the Caprifoliaceae [9] and share morphological characteristics, including obvious adaptations for the manipulation of the large pollen grains of that plant family, as listed below. It should be noted that molecular sequence data are available only for a part of the species and we therefore refrain from referring to the group as monophyletic, although it likely is.

**Female:** scopal hairs finely and densely plumose, with numerous short branches that often diverge from axis of, and nearly attain apex of main rachis to distance shorter than length of long hair branch; stipital comb spacing wide, with interspaces between adjacent comb teeth more than basal tooth width to about two basal tooth widths; maxillary palpus 5- or 6-segmented, the apical two segments smallest and weakly differentiated in some specimens (but 4- or 5-segmented with minute apical segment in *T. eoacinctella* **sp. nov.**), and segments 2 and 3 slender and longest, 5–6 times as long as wide; sternites 2–5 with plumose finely branched hairs (as seen in high magnification).

In females of common *Tetralonia* species with Asteraceae host plants, the scopal hairs are typically more coarsely branched, with few, comparatively long hair branches that are oriented along the same axis of the main rachis, and are far from its apex to a distance about equal to that of hair branches or much longer. The stipital comb is much denser, with the maximal interspaces between adjacent comb teeth being one basal diameter of a tooth or smaller. The maxillary palpus usually has six well defined segments, and either segments 2 and 3 or both are shorter, not much longer than segment 4, about four times as long as wide or less. Sternites 2–5 have mostly stiff, unbranched hairs, except for branched hairs along the apical margins. For example, the species *T. vicina* Morawitz, 1876 [15], which was considered as conspecific with *T. scabiosae* Moc. [16] (p. 169), is indeed reminiscent of the *pollinosa*-group in the surface sculpture of the mesonotum and T1 (as well as S6 of the male), but it agrees with the diagnosis listed above (see also in [17] (p. 149), including the illustration of the contrasting structures of scopal hairs).

**Male:** gonostylus elbowed, bent ventroapically in lateral view (Figure 1c), with blunt to sharp medial angle on inner side and with apex weakly to strongly expanded medially, as seen in dorsal view (Figures 2a,f,k,p and 3a,e); S8 with posterior emargination comparatively shallow, with short apical lobes or apical margin almost transverse; lateral process of S7 undivided into two lobes, the sclerotized posterior lobe reduced to mere apicolateral point (Figure 2c,h,m,r), and in two species, *T. cinctella* (Saund.) and *T. eoacinctella* **sp. nov.**, the lobes further modified to enclose deep ventral concavity (Figure 3c,g); posteromedial process of S7 produced with posteriorly projecting process (Figure 2c,h,m,r), but simply linear in *T. eoacinctella* **sp. nov.** (Figure 3g) and inconspicuous in *T. cinctella* (Saund.) (Figure 3c); S6 with posterolateral carinae conspicuous (except short and inconspicuous in *T. strigata* (Lep.), Figure 2i), curved anterolaterally, and, in some species, with anteriorly converging carina (Figures 2d and 3d,h), but in others with weakly defined to indistinct shallow ridge (Figure 2n,s).

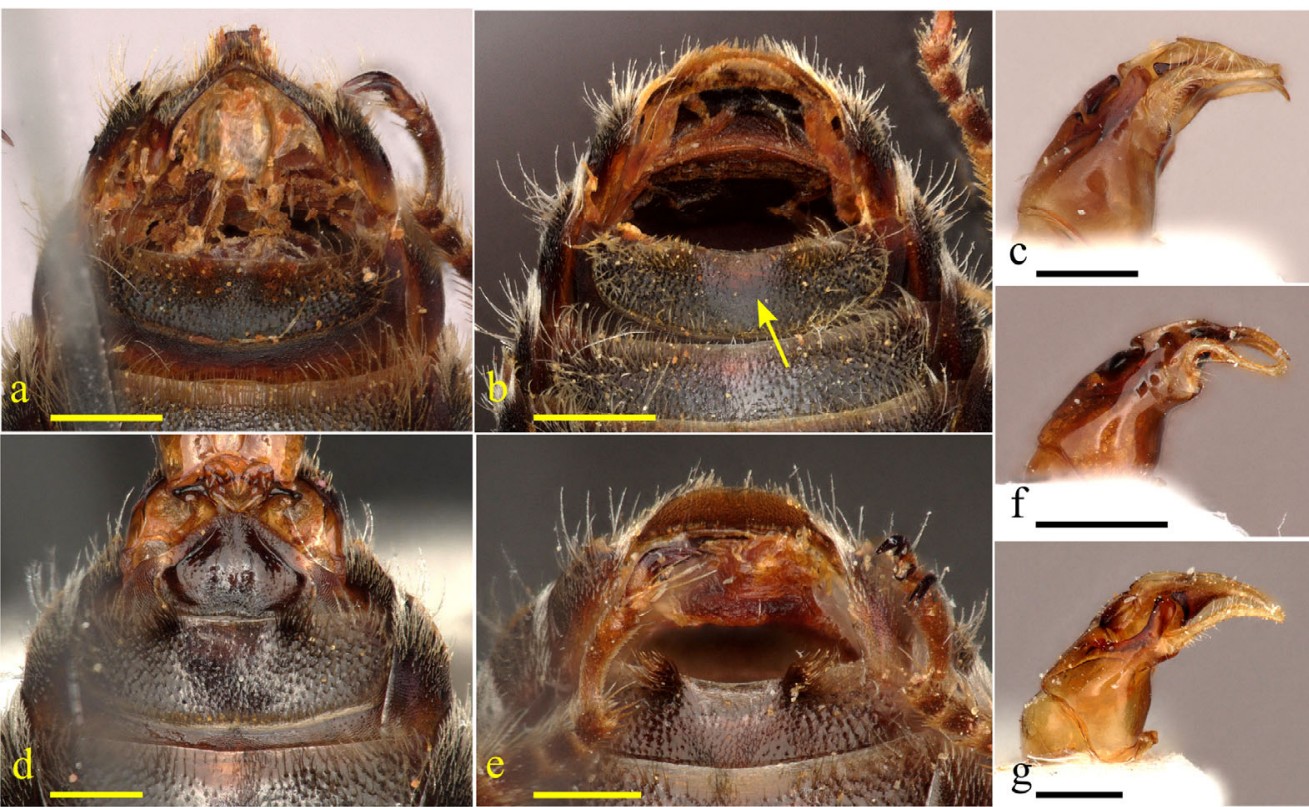

**Figure 1.** Males' sternite 5, and profile of genitalia. (**a**) *Tetralonia eoacinctella* **sp. nov.**; (**b**,**c**) *T. stellipilis* **sp. nov.**, arrow indicating depressed smooth apicomedial region of S5; (**d**) *T. nana* Mor., 1874; (**e**,**f**) *T. epilobii* **sp. nov.**; and (**g**) *T. malvae* (Rossi, 1790). Scale bars are 0.5 mm. Collection data of type specimens: paratypes, Israel and Palestine, (**a**) 1.1 km NW Kefar HaNasi, 246 m, 32.9855° N/35.5958° E, 23.v.2011, A. Dorchin leg.; (**b**,**c**) Mt. Hermon, 2180 m, 33.3155° N/35.8077° E, 21.viii.2012, at *Cephalaria stellipilis* Boiss., A. Dorchin leg.; and (**e**,**f**) 'En A Tina, 71 m, 33.0783° N/35.6443° E, 29.vii.2011, at *Lythrum salicaria* L., A. Dorchin leg.

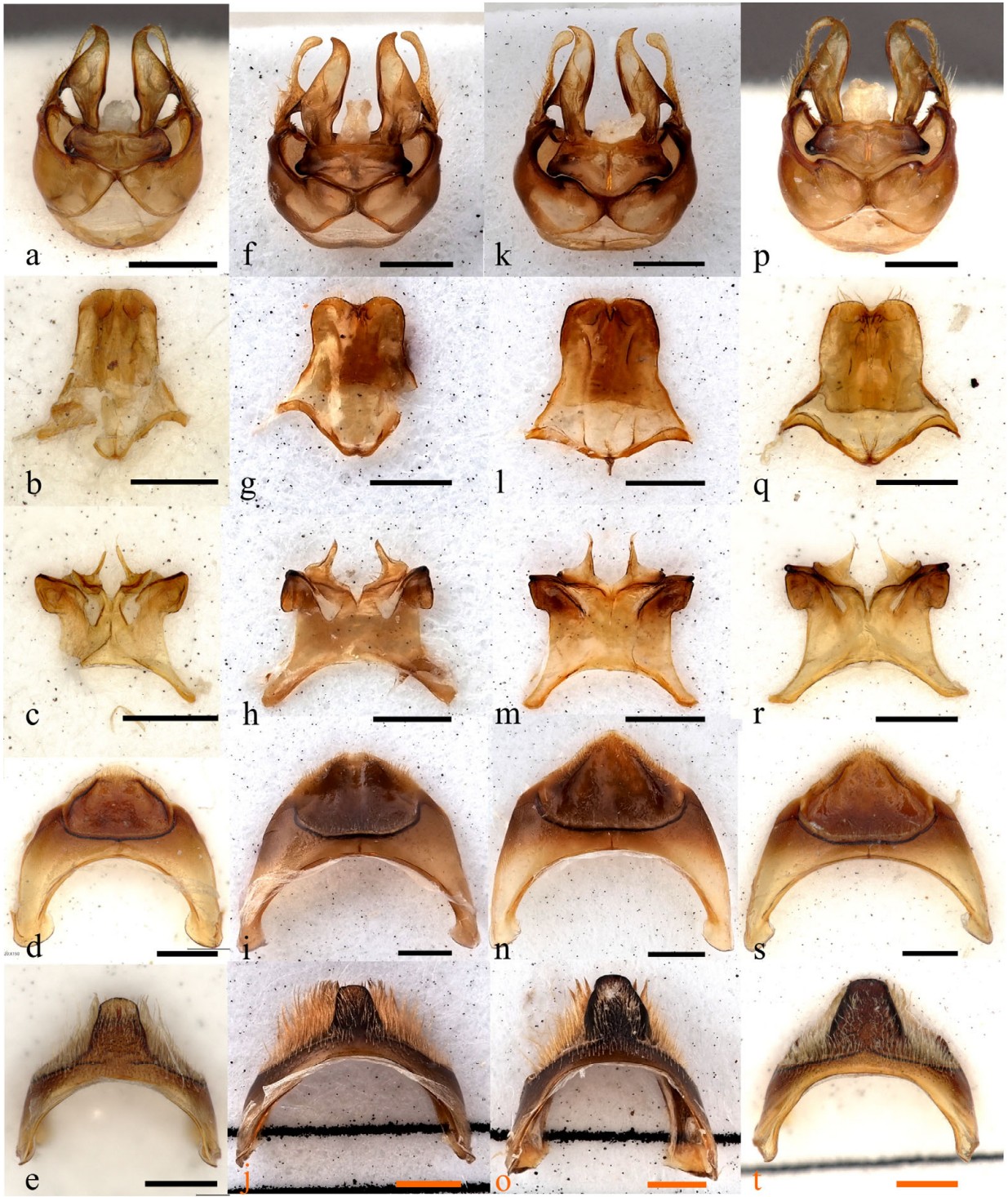

**Figure 2.** Male genitalia, S6–8 and T7 of the *pollinosa*-group. (**a**–**e**) *Tetralonia scabiosae* Moc., 1879; (**f**–**j**) *T. strigata* (Lep., 1841); (**k**–**o**) *T. pollinosa* (Lep., 1841); and (**p**–**t**) *T. stellipilis* **sp. nov**. Scale bars are 0.5 mm. Collection data of type specimen: (**p**–**t**) paratype, Israel and Palestine, Mt. Hermon, 2180 m, 33.3155° N/35.8077° E, 21.viii.2012, at *Cephalaria stellipilis* Boiss., A. Dorchin leg.

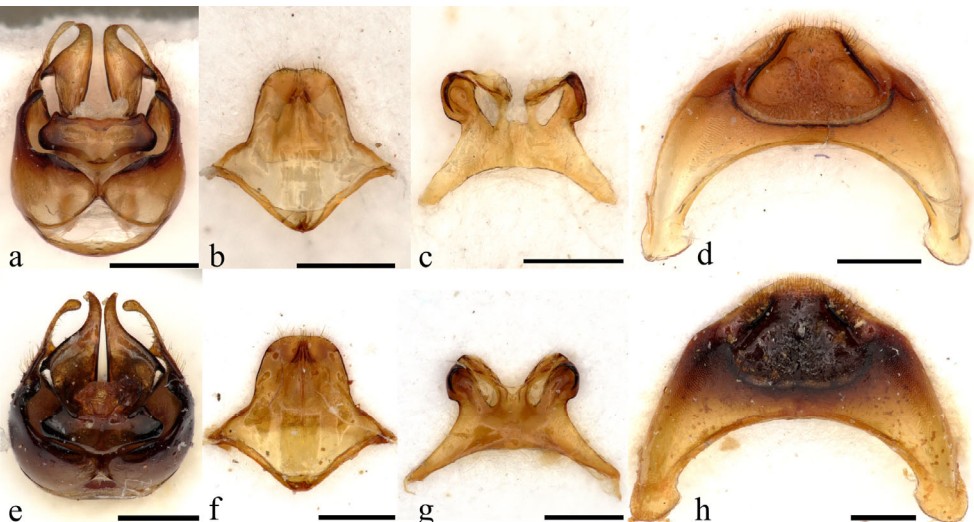

**Figure 3.** Male genitalia and S6–8 of the *pollinosa*-group (continued). (**a**–**d**) *Tetralonia cinctella* (Saund., 1908) and (**e**–**h**) *T. eoacinctella* **sp. nov**. Scale bars are 0.5 mm. Collection data of type specimen: (**e**–**h**) paratype, Israel and Palestine, 1.1 km NW Kefar HaNasi, 246 m, 32.9855° N/35.5958° E, 23.v.2011, A. Dorchin leg.

In the males of species outside the *pollinosa*-group, the lateral process of S7 is usually divided into two lobes by deep emargination, but it is undivided, as described above, in some unrelated species such as *T. dentata* (Klug, 1835) [18] and the *nana*-group of species described below. Other characteristics mentioned as diagnostic, such as the structures of the gonostylus, S8 and S6 are varying in extent in the genus *Tetralonia* as a whole, but S8 is frequently more deeply emarginated (including in the species just mentioned). In addition, the differences in the structure of the maxillary stipes and palpal segments that were mentioned in the female's account are applicable also to males.

3.1.2. Key to *pollinosa*-Group Species

1. Females: hind tibia and basitarsus with stiff scopal hairs; antenna short, 12-segmented; metasoma with six visible tergites ........................................**2**
-. Males: hind tibia and basitarsus without scopa; antenna long, 13-segmented; metasoma with seven visible tergites ........................................**7**
2. (1). Comparatively large, body length 10.5–12 mm, forewing length 8–9 mm; mesonotum with posteromedial region sparsely or less conspicuously punctate than laterally, with interspaces several puncture diameters wide, or surface scarcely to largely impunctate (Figure 4a–c); T1 with weakly defined, shallow asymmetric punctures anteromedially, and with marginal zone largely impunctate medially, except for some small to minute infiltrating punctures, thus impunctate surface wide, equal in width to about 1/3–1/2 of length of main disc (Figure 4g–i,m–o) .....................**3**
-. Comparably large or smaller, body length 9–11 mm, forewing length 7–8 mm; mesonotum with deep conspicuous punctures posteromedially, at most smaller and sparser than laterally, with only few interspaces more than three punctures wide (Figure 4d–f); T1 with well-defined, deep punctures anteromedially, and with comparatively narrow impunctate surface on marginal zone, equal in width to <1/3 length of main disc medially, although sometimes boundary of impunctate apical region inconspicuous due to sparse and/or minute punctures posteromedialy (Figure 4j–l) ........................................**5**
3. (2). T2 with main disc sparsely punctate medially, with comparatively well-defined large punctures, and interspaces more than two puncture diameters wide and often much wider, strongly differing from more densely punctate marginal zone (Figure 4i,o); T2–4 with dark unbranched fine simple hairs, uncovering surface of main discs, and

with white apical fasciae covering margins, widely interrupted on T2, complete and entirely concealing margin of T4 in fresh specimens (Figure 4i); prepygidial and pygidial fimbria dark brown to black .............. *Tetralonia strigata* (**Lepeletier, 1841**)

-. T2 with comparably dense, small punctures both medially on main disc and marginal zone, with most interspaces up to two punctures wide (Figure 4h,n), or punctures on marginal zone faint and weakly identifiable (Figure 4g,m); T2–4 with light basal tomentum, progressively covering greater portions of tergites, and with marginal zones widely to narrowly exposed medially, that of T4 briefly or inconspicuously so even in fresh specimens (Figure 4g,h); prepygidial and pygidial fimbria bright fulvous (Figure 4h) to light brown ................................. **4**

4. (3). Comparatively large, body length 12 mm, forewing length 9 mm; T2 with recumbent tomentum limited to base, not attaining marginal zone even in fresh specimens, there with marginally identifiable scarce punctures medially (as on immediately adjacent posterior of main disc) (Figure 4g,m); mesonotum comparatively densely punctate, with interspaces up to puncture diameter wide laterally near parapsidial line, and posteromedial region narrowly impunctate (Figure 4a); vestiture with some darker brown hairs on mesonotum and scutellum (Figure 4a), most of T5 and pygidial fimbria of T6 ........ *Tetralonia pollinosa* (**Lepeletier, 1841**)

-. Comparatively small, body length 10.5–11 mm, forewing length 8 mm; T2 with recumbent tomentum expanded onto marginal zone except medially, there with small distinct dense punctures, with most interspaces up to two punctures wide (comparable to immediately adjacent posterior of main disc) (Figure 4h,n); mesonotum comparatively sparsely punctate, with many interspaces two puncture diameters wide or more laterally near parapsidial line, and posteromedial region widely impunctate (Figure 4b); vestiture uniformly pale greyish fulvous, bright fulvous medially on T5 and on pygidial fimbria (Figure 4b,h) ................................. *Tetralonia stellipilis* **sp. nov.**

5. (2). Mesonotum densely punctate with no visible interspaces laterally near parapsidial line, and most interspaces less than puncture diameter on posteromedial region with some confluent punctures (Figure 4d); T1 with dense large punctures posteromedially, with interspaces 1–2 puncture diameters wide on premarginal area, and mostly less than puncture diameter on marginal zone (Figure 4j); T2–5 with complete bands of white apical fasciae in fresh specimens (attenuate medially on T2, mixed with golden hairs medially on T5) and sparse unbranched dark hairs that expose anterior rest of tergites, except for narrow, often unexposed, white basal tomentum on T2 (Figure 4j) .............. *Tetralonia cinctella* (**Saunders, 1908**)

-. Mesonotum more sparsely punctate, with some puncture wide interspaces laterally near parapsidial line, and with many interspaces 1–2 puncture diameters wide on posteromedial region (Figure 4e,f); T1 with sparse, small or minute punctures posteromedially, with interspaces multiple puncture diameters (Figure 4k,l); T2–4 with anterior as well as posterior fulvous tomentum, progressively increasing in extent and entirely covering T4, and T5 with ferruginous hairs (Figure 4k,l) ................................................................. **6**

6. (5). Comparatively small, body length 9–9.5 mm, forewing length 7 mm; T1 comparatively sparsely punctate, with puncture density on anteromedial half of tergite comparable to posteromedial half, with most interspaces more than two puncture diameters wide and often much wider (Figure 4l); T2 with apical fasciae widely interrupted, that of T3 complete, inconspicuously exposing margin medially even in fresh specimens (Figure 4l) .................... *Tetralonia scabiosae* **Mocsàry, 1879**

-. Comparatively large, body length 10.5–11 mm, forewing length 7.5–8 mm; T1 more densely punctate on anteromedial half than posteromedially, with most interspaces less than two puncture diameters wide, although with some much wider interspaces (Figure 4k); T2, as T3, with complete apical fasciae extended well over, and entirely covering margin in fresh specimens (Figure 4k) ....... *Tetralonia eoacinctella* **sp. nov.**

7. (1). Clypeus and labrum with broad light maculation (cf. Figure 6e); S5 with apicomedial region depressed, smooth, weakly punctate and largely hairless, with hairs on both sides sparse enough to expose underlain surface, not formed into conspicuous tufts (Figure 1b) ............................................................................................................ **8**

-. Clypeus and labrum dark, immaculate (cf. Figure 5e); S5 with apicomedial region not conspicuously depressed, with some hairs and sparse minute punctures, and with dense tufts of hairs on both sides that hide underlain surface (Figure 1a) ............................................................................................................ **11**

8. (7). Small, body length 9.5–10 mm, forewing length 6.5–7 mm; mesonotum with well-defined deep punctures on posteromedial region (cf. Figure 4f); penis valves abruptly tapering on curved apex, as seen in dorsal view (Figure 2a); T2–5 with recumbent tomentum that completely hide anterior underlain surface in fresh specimens ............................................................ *Tetralonia scabiosae* **Mocsàry, 1879**

-. Comparatively large, body length 10.5–12 mm, forewing length 7.5–8 mm; mesonotum with widely impunctate posteromedial region or with weakly defined shallow punctures (cf. Figure 4a–c); penis valves gradually tapering apically, as seen in dorsal view (Figure 2f,k,p); T2–5 with semierect hairs, not completely hiding underlain anterior surface in fresh specimens (Figure 6d) ......................................... **9**

9. (8). S6 with only weak indication of posterolateral carina apically (Figure 2i); posteromedial region of mesonotum with faint, weakly defined punctures, at most with small impunctate area a few punctures wide; clypeus with light maculation reduced, bluntly subtriangular; T2–5 with dense apical fasciae that strongly differentiate from sparser hairs on anterior rest of tergites, and completely covering apical margins in fresh specimens, or almost so ........................... *Tetralonia strigata* (**Lepeletier, 1841**)

-. S6 with posterolateral carina conspicuous (Figure 2n,s); posteromedial region of mesonotum with a large impunctate area, multiple punctures wide; clypeus with light maculation large, sharply plus-shaped (Figure 6e); T2–5 uniformly covered with hairs, or hairs sometimes form denser premarginal bands, with margins of T2–4 exposed at least medially, even in fresh specimens (Figure 6d) ........................ **10**

10. (9). Gonostylus curvature comparatively weak such that apex weakly expanded medially, spatulate, and medial angle on inner side obtuse, as seen in dorsal view (Figure 2k); penis valve with lateral spine oriented more laterally, such that its lateral margin weakly basally concave, then convex (Figure 2k); T2 sparsely punctate, with most interspaces more than puncture diameter medially, as on T1 (cf. Figure 4g,m), easily observed due to lack of recumbent tomentum posteromedially ............................................ *Tetralonia pollinosa* (**Lepeletier, 1841**)

-. Gonostylus elbowed, with apex slender and strongly expanded medially, L-shaped (Figure 2p), and with right medial angle on inner side, as seen in dorsal view; penis valve with lateral spine oriented more basally, with lateral margin basally straight, then convex (Figure 2p); T2 with punctures denser than on T1, with most interspaces less than puncture diameter medially (cf. Figure 4h,n), entirely covered by recumbent tomentum in fresh specimens ............................ *Tetralonia stellipilis* **sp. nov.**

11. (7). T1 with well-defined, dense, deep punctures posteromedially, with most interspaces less than puncture diameter wide (cf. Figure 4j), delineating narrow, impunctate apical margin, 4–5 nearby punctures wide; first flagellar segment comparatively long, 1/4–1/5 as long as second; gonostylus curvature slightly weaker, with apex weakly expanded medially, spatulate (Figure 3a), and right medial angle on inner side, as seen in dorsal view; T3–5 with pale apical fasciae largely restricted to marginal zones and delimited from darker anterior rest of tergites, at most with sparse similar light hairs anteriorly (cf. Figure 4j) ............................ *Tetralonia cinctella* (**Saunders, 1908**)

-. T1 with shallow, weakly defined, sparse punctures posteromedially, with abundant interspaces one puncture wide (cf. Figure 4k), delineating broad impunctate apical margin, >6 nearby punctures wide; first flagellar segment comparatively short, ~1/6 as long as second; gonostylus curvature slightly stronger, with apex strongly expanded

mesally, L-shaped (Figure 3e), and sharp medial angle on inner side as seen in dorsal view; T3–5 with ferruginous apical fasciae extended anteriorly by similar shorter hairs on main disc in fresh specimens (cf. Figure 4k) ............. *Tetralonia eoacinctella* **sp. nov.**

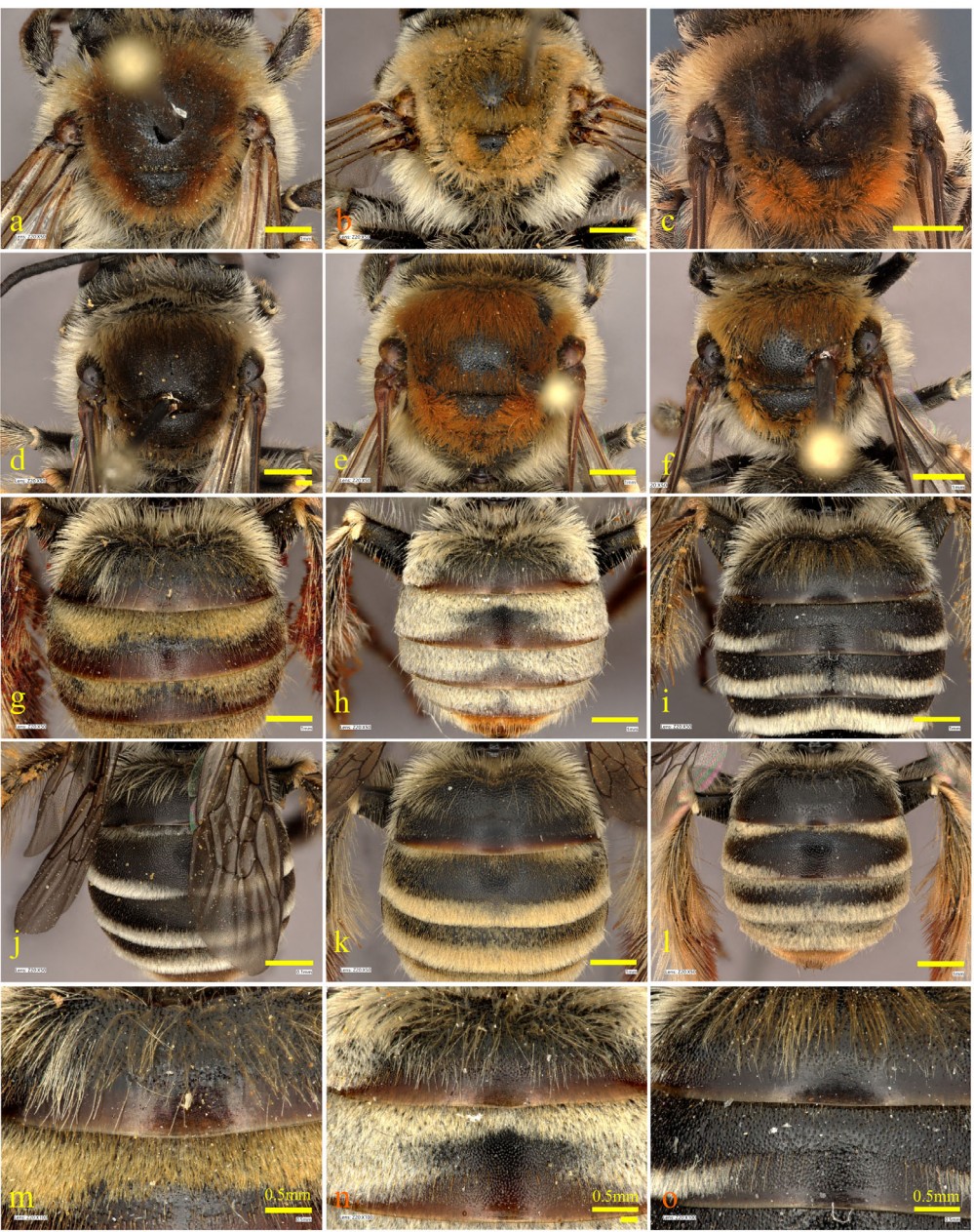

**Figure 4.** Mesosoma and metasoma of females of the *pollinosa*-group. Mesosoma of (**a**) *Tetralonia pollinosa* (Lep., 1841); (**b**) *T. stellipilis* **sp. nov.**; (**c**) *T. strigata* (Lep., 1841); (**d**) *T. cinctella* (Saund., 1908); (**e**) *T. eoacinctella* **sp. nov.**; and (**f**) *T. scabiosae* Moc., 1879. Metasoma of (**g**) *Tetralonia pollinosa* (Lep., 1841); (**h**) *T. stellipilis* **sp. nov.**; (**i**) *T. strigata* (Lep., 1841); (**j**) *T. cinctella* (Saund., 1908); (**k**) *T. eoacinctella* **sp. nov.;** and (**l**) *T. scabiosae* Moc., 1879. T1 and T2 of (**m**) *Tetralonia pollinosa* (Lep., 1841); (**n**) *T. stellipilis* **sp. nov.**; and (**o**) *T. strigata* (Lep., 1841). Scale bars are 1 mm, except when labeled 0.5 mm. Collection data of type specimens: paratypes, Israel and Palestine, (**b,h,n**) Mt. Hermon, 1600 m, 23.vii.2016, at *Cephalaria stellipilis* Boiss., A. Dorchin leg.; and (**e,k**) Mt. Gilboa', N Merav, 353 m, 32.4575° N/35.4244° E, 14.v.2017, at *Cephalaria joppensis* (Rchb.) Coult., A. Dorchin leg.

3.1.3. Taxonomy

***Tetralonia cinctella* (Saunders, 1908)**

*Eucera cinctella* Saunders, 1908 [14] (p. 258). ♀♂, "La Calle" (♀♂), "Tizi Ouzou, Mount Beloua" (♀) (Algeria). Lectotype: ♀, "La Calle" (Algeria), BMNH, designated here.

The female lectotype is the only syntype that was found in BMNH, with no trace for the remaining type series. The lectotype was examined with the help of the curator J Monks, it agrees with the original description, and was confirmed as conspecific with the recent material studied in this work.

***Tetralonia pollinosa* (Lepeletier, 1841)**

*Macrocera pollinosa* Lepeletier, 1841 [12] (p. 92). ♀(partim), "Environs de St.-Sever. Envoyée par le savant M. Léon Dufour.". Lectotype: ♀, OUM, designated by Baker [19] (p. 1200).

*Tetralonia mediocris* Eversmann, 1852 [20] (p. 122). ♀♂, "in promontoriis Uralensibus Australibus" (Orenburg province, Russia). Lectotype: ♀, "Spask., Jul." (Spasskoe, Orenburg Province, Russia), ISEAP, designated by Proshchalykin et al. [21] (p. 35). Synonymy in Dalla Torre [22] (p. 244), Friese [23] (p. 72).

*Tetralonia canescens* Dours, 1873 [24] (p. 325) (originally given as *Tetralonia canescens* L. Duf.). ♂, "Alg. Esp. Fr. mérid.". Type material presumed lost [25] (p. 60). Synonymy in Pérez [26] (p. 151).

*Tetralonia fossulata* Morawitz, 1874 [27] (p. 142). ♂, "Derbent" (Dagestan Republic, Russia). Syntypes probably lost, not found in ZINSP and ISEAP [27]. Synonymy in Dalla Torre [22] (p. 244), Friese [23] (p. 72).

?*Tetralonia adusta* Mocsàry, 1877 [28] (p. 233). ♀, "In Hungária centrali". Syntype?: "Tasnád" (Tășnad, northern Romania), MTM. Synonymy in Dalla Torre [22] (p. 244), Friese [23] (p. 72).

***Tetralonia scabiosae* Mocsàry, 1879**

*Tetralonia scabiosae* Mocsàry, 1879 [13] (p. 21). ♀♂, "In Hungaria meridionali ad Grebenácz et Jassenovam" (from Grebenac to Jasenovo, Serbia). Lectotype: ♂, MTM, designated by Tkalců [17] (p. 148).

***Tetralonia strigata* (Lepeletier, 1841)**

*Macrocera strigata* Lepeletier, 1841 [12] (p. 104). ♂, "Espagne". Type material presumed lost [19] (p. 1197).

*Macrocera subundulata* Lepeletier, 1841 [12] (p. 106). ♀, "Espange". Lectotype, OUM, designated by Baker [19] (p. 1198). Synonymy in Pérez [29] (p. XLV), [given as "*strigata* Friese" = *Eucera strigata* (Lepeletier), Friese [23] (p. 96)].

*Tetralonia inoequidistans* Dours, 1873 [24] (p. 324). [Originally given as "*Tetralonia* (*Macrocera*) *inoequidistans* L. Duf.", lapsus]. (♂), "Esp. Alg. France mérid. Perpign. Coll. L. Duf. Dours." (Spain; Algeria; Perpignan, Southern France). Type material unlocated, not found in MNHN. Synonymy in Pérez [26] (p. 155) as "Macrocera inæquidistans Duf." (see below).

*Macrocera inaequidistans* Dufour, Pérez [26] (p. 155). ♂, "Perpignan" (Perpignan, Southern France) (partim). Type material unlocated, not found in MNHN. Synonymy in Pérez [26] (p. 155).

*Macrocera antigae* Pérez, 1902 [29] (p. XLV). ♀♂, "Barcelone". Type material unlocated, not found in MNHN. Synonymy in Dusmet y Alonso [30] (p. 165).

? *Macrocera confusa* Pérez, 1902 [29] (p. XLVI). ♂, "Andalousie". **Syn. nov.** Type material unlocated, not found in MNHN.

***Macrocera inaequidistans* Dufour, Pérez** [26]

Pérez [26] (p. 155) studied two male specimens under this name in coll. Dufour, which he received from A Laboulbéne. These were the same type specimens examined by Dours [24] that should have remained in Dufour's collection, now preserved in MNHN, but they were not found there. Pérez [26] mentioned that they were not conspecific, the one from Perpignan, France, was conspecific with *Tetralonia strigata*, and the other from Spain was probably conspecific with *Tetralonia mediocris* Eversmann, 1852 = *Tetralonia pollinosa* (Lepeletier, 1841).

*Macrocera antigae* **Pérez, 1902**

Dusmet y Alonso [30] (p. 166) examined male and a female syntypes in MNHN, and the description he provided fits both sexes of *Tetralonia strigata* (Lepeletier, 1841) well. He suspected that they were conspecific with that species, but he could not reach a decisive conclusion, since the Lepeletier types were no longer available to him. Baker [19] (p. 1198) studied the type material of *Macrocera subundulata* Lepeletier, 1841 [12] in OUM, which was conspecific [29], and has also put the synonymy in question because of not seeing the types of *Macrocera antigae*.

*Macrocera confusa* **Pérez, 1902**

Dusmet y Alonso [30] (p. 173) examined a type in MNHN and suspected it of being a mere variation of *Macrocera antigae* Pérez, 1902 [29], with most distinguishing characteristics being weak, 'except for the lack of lateral tufts of hairs on S5', which he had not seen in other specimens ("la falta de los mechones laterales de pelos en el quinto segmento ventral no lo he observado en ninguno de mis numerosos ejemplares"), but these hairs are actually comparatively sparse in that species, not forming conspicuous tufts.

3.1.4. Description of New Species

*Tetralonia eoacinctella* **Dorchin sp. nov.**

**Holotype:** ♂, ISRAEL AND PALESTINE, Mt. Gilboa', N Merav, 353 m, 32.4575° N/35.4244° E, 14.v.2017, at *Cephalaria joppensis* (Rchb.) Coult., A. Dorchin leg. (SMNH).

**Paratypes:** ISRAEL AND PALESTINE: 6♂, 1.1 km NW Kefar HaNasi, 246 m, 32.9855° N/35.5958° E, 23.v.2011, A. Dorchin leg. (♂OLML, ♂SMNH, 4♂ADCM); 2♂7♀, SW Nir Dawid, −107 m, 32.5019° N/35.4491° E, 14.v.2017, at *Cephalaria joppensis* (Rchb.) Coult., A. Dorchin leg. (2♀OLML, ♀SMNH, 2♂4♀ADCM); 7♂, Mt. Gilboa', N Merav, 353 m, 32.4575° N/35.4244° E, 14.v.2017, at *Cephalaria joppensis* (Rchb.) Coult., A. Dorchin leg. (2♂OLML, ♂SMNH, 4♂ADCM).

**Additional material:** ISRAEL AND PALESTINE: ♀, Sha'alvim, 27.v.2010, G. Pisanty leg.; Tal Shahar, 23.v.2020, at *Cephalaria joppensis* (Rchb.) Coult., T. Roth leg.; ♂, Nahshon, 29.v.2020, at *Cephalaria joppensis* (Rchb.) Coult., T. Roth leg.; ♂4♀, Nahshon, 1.vi.2020, at *Cephalaria joppensis* (Rchb.) Coult., T. Roth leg.; ♂, Latrun, 25.v.2020, at *Cephalaria joppensis* (Rchb.) Coult., T. Roth leg.; ♂, Hulda, 2.vi.2020, at *Cephalaria joppensis* (Rchb.) Coult., T. Roth leg.

**Diagnosis and description:** As implied by its name, this species is most reminiscent of the western Mediterranean species *T. cinctella* (Saunders). The species agrees with the characteristics listed in the diagnosis for the group, and shares with *T. cinctella* the specific character variations that were mentioned there. These, and additional characteristics that are provided in the key, are only briefly mentioned below.

**Female**: Body length 10.5–11 mm, forewing length 7.5–8 mm; integument overall dark, with margin of T1 uncovered by hairs and widely translucent; vestiture cream white, ferruginous dorsally on mesosoma and pygidial and prepygidial fimbria; T2–4 with anterior as well as posterior bands of fulvous tomentum, progressively increasing in extent, entirely covering T4, extended well over apical margins, and entirely covering margins in fresh specimens (Figure 4k, Figure 5a,c); scopal hairs cream white, with multiple fine branches (cf. Figure 7a), dark brownish ferruginous on ventral side of basitarsus; stipital teeth moderately sparse, teeth interspaces >1×, <2× basal tooth width (as in *T. cinctella*, and slightly narrower than in *T. pollinosa* or *T. strigata*); maxillary palpus 4- or 5-segmented with minute apical segment; mesonotum with deep conspicuous punctures, with some puncture wide interspaces laterally near parapsidial line, and with most interspaces 1–2 puncture diameters wide on posteromedial region (but with some wider interspaces, and overall slightly sparser than in *T. cinctella*) (compare Figure 4e to Figure 4j); T1 with well-defined, deep punctures anteromedially (compared with weakly defined asymmetric punctures in *T. pollinosa* or *T. strigata*), comparatively sparse medially on main disc, with interspaces multiple puncture diameters; the punctures become scarce, irregular, and small posteriorly, with minute punctures on anterior of marginal zone, leaving comparatively narrow impunctate apical surface (although boundary of impunctate apical surface inconspicuous) (Figure 4k); T2

with well-defined punctures on main disc, shallower and sparser medially with numerous interspaces ≥2 puncture diameters (compared with mostly <2 punctures wide in *T. cinctella*) (Figure 4k).

**Male**: Body length 10–11 mm, forewing length 7–8 mm; integument overall dark as in female; vestiture dark to light ferruginous on vertex, mesonotum and tergites, with complete apical fasciae that cover margins of T2–5, and extended anteriorly by similar shorter hairs onto main discs of T3–5 in fresh specimens (cf. Figure 4k) (these hairs much sparser, such that apical fasciae not visibly extended anteriorly in *T. cinctella*); clypeus and labrum dark, immaculate (Figure 5e); first flagellar segment comparatively short, ~1/6 as long as second (shorter than in *T. cinctella*); maxillary palpus as in female, and stipital comb spacing slightly wider, up to 2 basal tooth diameters; mesonotum with deep dense punctures, as in female; T1 weakly and shallowly punctate posteromedially (cf. Figure 4k), and with comparatively broad impunctate apical margin (punctation more conspicuous and abundant in *T. cinctella* as described in the key); T2 with punctures distinctly smaller and denser than on T1, with numerous interspaces puncture diameter or more medially on main disc (thus slightly sparser than in *T. cinctella*); S5 without differentiated apicomedial region, with dense submedial tufts of hairs (Figure 1a); S6 with both conspicuous posterolateral carinae and short anterior carina or ridge, with each portion of the converging carinae distinct (not continuous as in *T. cinctella*) (compare Figure 3h to Figure 3d); pygidial plate slender, as in *T. strigata* (cf. Figure 2j), or broader; gonostylus elbowed, as seen in lateral view (cf. Figure 1c), with apex strongly expanded medially, L-shaped (Figure 3e), and with sharp medial angle on inner side as seen in dorsal view; S8 short and broad, with apical margin transverse (Figure 3f); lateral process of S7 undivided into two lobes, enclosed into deep ventral concavity (Figure 3g); posteromedial process of S7 simply linear (Figure 3g).

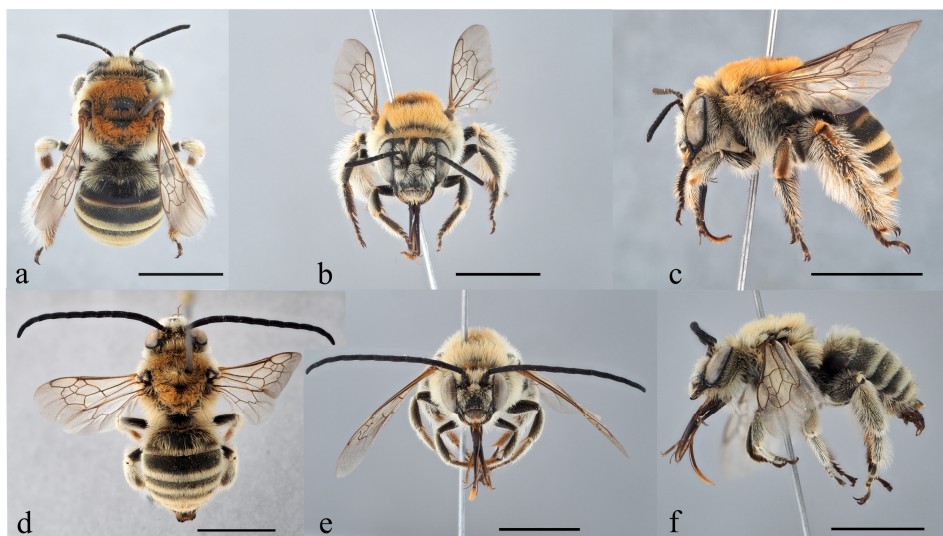

**Figure 5.** Female and male habitus of *Tetralonia eoacinctella* **sp. nov**. (**a**–**c**) Female in dorsal, frontal, and lateral view and (**d**–**f**) male holotype in dorsal, frontal, and lateral view. Scale bars are 5 mm. Collection data of type specimens: Israel and Palestine, (**a**–**c**) paratype, SW Nir Dawid, −107 m, 32.5019° N/ 35.4491° E, 14.v.2017, at *Cephalaria joppensis* (Rchb.) Coult., A. Dorchin leg.; and (**d**–**f**) holotype, Mt. Gilboa', N Merav, 353 m, 32.4575° N/35.4244° E, 14.v.2017, at *Cephalaria joppensis* (Rchb.) Coult., A. Dorchin leg.

**Etymology:** The new species name is a combination of the adjective "eoa" in Latin, meaning "eastern", and the species name "cinctella", to indicate the occurrence of the species in the Eastern Mediterranean region and its close morphological similarity to the Western Mediterranean species *T. cinctella* (Saunders).

**Distribution:** The new species is known from the north of Israel and Palestine.

*Tetralonia stellipilis* Dorchin sp. nov.

**Holotype:** ♂, ISRAEL AND PALESTINE, Mt. Hermon, 2180 m, 33.3155° N/35.8077° E, 21.viii.2012, at *Cephalaria stellipilis* Boiss., A. Dorchin leg. (SMNH).

**Paratypes:** ISRAEL AND PALESTINE: 2♂, Mt. Hermon, 1969 m, 21.viii.2011, at *Cephalaria stellipilis* Boiss., D. Furth leg. (2♂ADCM); 11♂7♀, Mt. Hermon, 2180 m, 33.3155° N/35.8077° E, 21.viii.2012, at *Cephalaria stellipilis* Boiss., A. Dorchin leg. (3♂2♀OLML, 2♂2♀SMNH, 6♂3♀ADCM); 2♀, Mt. Hermon, 1600 m, 23.vii.2016, at *Cephalaria stellipilis* Boiss., A. Dorchin leg. (2♀ADCM). LEBANON: ♂, Mont Liban, Kafraiyda, Maaser el Chouf—Kefraya Passage WGS84, 33.6708° N/35.7022° E, 6.viii.2018, at *Cephalaria stellipilis* Boiss., X. Van Achter leg (♂ADCM); ♀, Mont Liban, Maaser-Kefraya, 1788 m, 33.6708° N/35.7023° E, 15.viii.2019, at *Cephalaria stellipilis* Boiss., M. Boustani (♀ADCM). TURKEY: ♂, W Hakkari, Tanin-Tanin-Pass, 2300–2600 m, 14.viii.1979, K. Warnke, leg. (♂ADCM).

**Diagnosis and description:** This species is most reminiscent of *T. pollinosa* (Lep.), with which it shares the characteristics listed in the group diagnosis as well as the key, which are only briefly mentioned below. This is the only species in the *pollinosa*-group for which molecular sequence data were provided in [2] as sample ad38.

**Female:** Body length 10.5–11 mm, forewing length 8 mm; integument overall dark, with tergite margins and entire of tegulae lighter and translucent (Figure 4b,h,n); vestiture uniformly pale greyish fulvous, bright fulvous medially on T5 and on pygidial fimbria (Figure 4b,h,n and Figure 5a–c); mesosoma with uniformly short, erect, finely branched hairs (Figure 4b), longer on vertex and on T1, and tergites progressively covered by light basal tomentum, with marginal zone of T2 widely exposed, and those of T3 and T4 narrowly and briefly exposed, respectively, in fresh specimens (Figure 4h,n); scopal hairs white, with multiple fine branches (cf. Figure 7a), dark brownish ferruginous on ventral side of basitarsus; mesonotum comparatively sparsely punctate, with widely impunctate postero-medial region (Figure 4b); T1 with weakly defined asymmetric punctures, denser than in *T. pollinosa*, with most interspaces less than two puncture diameters anteromedially, and with marginal zone largely impunctate for most length, except for minute infiltrating punctures medially (compare Figure 4n to Figure 4m); T2 and T3 with distinct small punctures on marginal zones as well as anteriorly, denser and more conspicuous than in *T. pollinosa*, with most interspaces up to two punctures wide (compare Figure 4h,n to Figure 4g,m); stipital comb spacing wide, with maximal interspaces two basal tooth widths; maxillary palpus typically 5-segmented, rarely 6-segmented with the apical two segments weakly differentiated, and segments 3 longest.

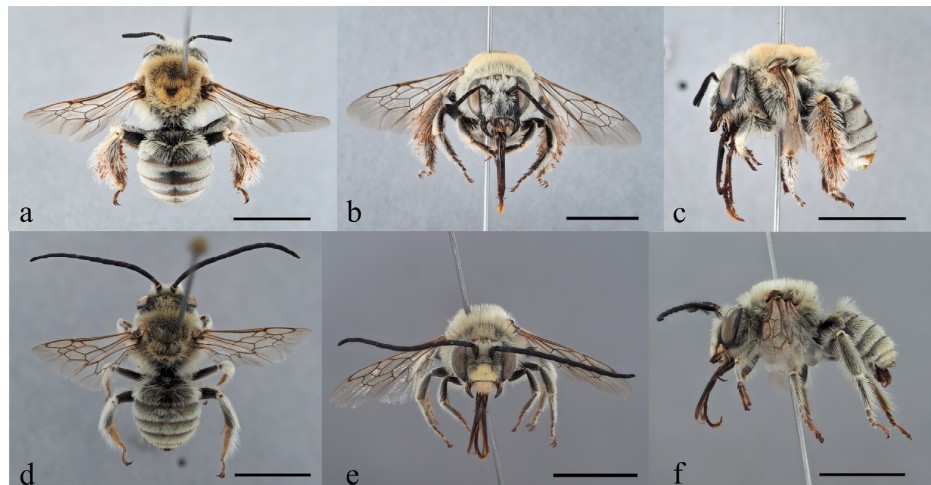

**Figure 6.** Female and male habitus of *Tetralonia stellipilis* **sp. nov**. (**a–c**) Female in dorsal, frontal, and lateral view and (**d–f**) male holotype in dorsal, frontal, and lateral view. Scale bars are 5 mm. Collection data of type specimens: Israel and Palestine, (**a–c**) paratype, Mt. Hermon, 1600 m, 23.vii.2016, at *Cephalaria stellipilis* Boiss., A. Dorchin leg.; and (**d–f**) holotype, Mt. Hermon, 2180 m, 33.3155° N/35.8077° E, 21.viii.2012, at *Cephalaria stellipilis* Boiss., A. Dorchin leg.

**Male:** Body length 10.5–11 mm forewing length 7.5 mm; clypeus with plus shape light yellow maculation, and sometimes supraclypeal area with small yellow spot (Figure 6e); labrum completely ivory white (Figure 6e); mandible entirely yellow at base (Figure 6f); integument overall dark and vestiture uniformly greyish fulvous, as in female (Figure 6d–f), darker fulvous in male from Hakkari, Turkey; tergites with semierect plumose hairs basally and T2–5 with recumbent tomentum posteriorly, not extended beyond margins of T2, T3, and middle of T4 (Figure 6d,f); T6 with light to dark (in Turkish specimen) fulvous hairs (Figure 2t); mesonotum with wide impunctate posteromedial area, as in female (cf. Figure 4b); T1 with sparse asymmetric punctures anteriorly, increasing in density posteriorly, denser than in *T. pollinosa*, and delineating more or less clear boundary with impunctate margin (despite being weaker in some specimens), with smooth margin comparatively narrow, not occupying entire of marginal zone; T2 with punctures denser than on T1 and entirely covered by recumbent tomentum, as given in the key; stipital comb widely spaced and maxillary palpus 5-segmented, as in female; gonostylus curvature strong, and lateral spine of penis valve oriented basally (Figure 2p), more so than in *T. pollinosa*, as described in the key; apical lobes of S8 rounded, divided by small shallow emargination (Figure 2q), but emargination indistinct in male from Hakkari, Turkey, as in *T. pollinosa*; S7 typical of the group (Figure 2r); S6 typical of the group, with conspicuous posterolateral carina, and only weak basal ridge, thus not visibly attaining basal portion of sternum (Figure 2s).

**Etymology:** The new species is named after its known host plant, *Cephalaria stellipilis* [9], and is given as a noun in apposition.

**Distribution:** The new species is restricted to the Levant region, and was recorded so far only from Mt. Hermon in Israel and Palestine, Mt. Lebanon, and one male was identified from the Hakkari region of Turkey.

*3.2. nana-Group*

3.2.1. Diagnosis

The group includes only two known species, *Tetralonia nana* Mor. and *T. epilobii* **sp. nov**. The male genital characteristics are reminiscent of those of the *pollinosa*-group, and particularly *T. scabiosae* Moc. (but not *T. malvae* (Rossi)). The female scopa is reminiscent of that of *T. malvae* in having particularly sparse scopal hairs (compare Figure 7b,c to Figure 7d), but this is obviously due to shared host plants, where *T. nana* specializes on Malvaceae, particularly the genus *Althaea* [16] (p. 162) (AD, pers. obs.). *T. epilobii* **sp. nov.** specializes on the genus *Epilobium* in the family Onagraceae, which has large pollen grains similar to those of Malvaceae [9]. Molecular sequence data were provided for this latter species in [2] as sample ad36, while *T. nana* was not included. The following diagnostic characteristics are shared between the two species.

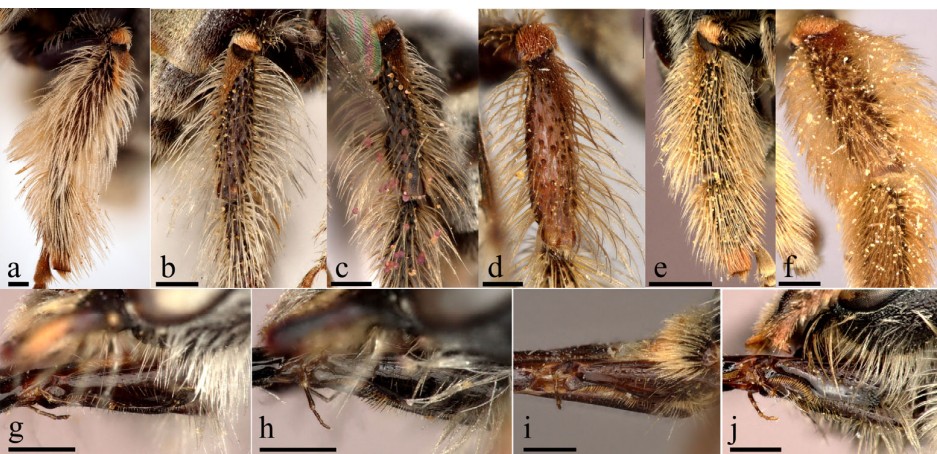

**Figure 7.** (**a–f**) Female scopa. (**a**) *Tetralonia scabiosae* Moc., 1879; (**b**) *T. epilobii* **sp. nov.**; (**c**) *T. nana* Mor., 1874; (**d**) *T. malvae* (Rossi, 1790); (**e**) *T. salicariae* (Lep., 1841); and (**f**) *T. glauca* (Fabricius, 1775). (**g–j**) Stipes

and maxillary palpus. (**g**) *T. epilobii* **sp. nov.**; (**h**) *T. nana* Mor., 1874; (**i**) *T. malvae* (Rossi, 1790); and (**j**) *T. salicariae* (Lep., 1841). Scale bars are 0.5 mm. Collection data of type specimen: (**b,g**) paratype, Israel and Palestine, 'En A Tina, 71 m, 33.0783° N/35.6443° E, 29.vii.2011, at *Lythrum salicaria* L., A. Dorchin leg.

**Female:** scopal hairs on dorsal side of both tibia and basitarsus sparse enough to uncover underlain surface, arising from tubercles that are spaced apart from closest tubercles by mostly more than one tubercle diameter (Figure 7b,c); anterior basitarsus and mediotarsal segments with elongated, curved, stiff unbranched hairs (Figure 8a,d); maxillary palpus 5- or 6-segmented, sometimes with two apical segments small and weakly differentiated (Figure 7g,h); mesonotum sparsely punctate with some smooth interspaces 1–2 punctures wide laterally near parapsidial line and wide surfaces multiple punctures wide on postero-medial region.

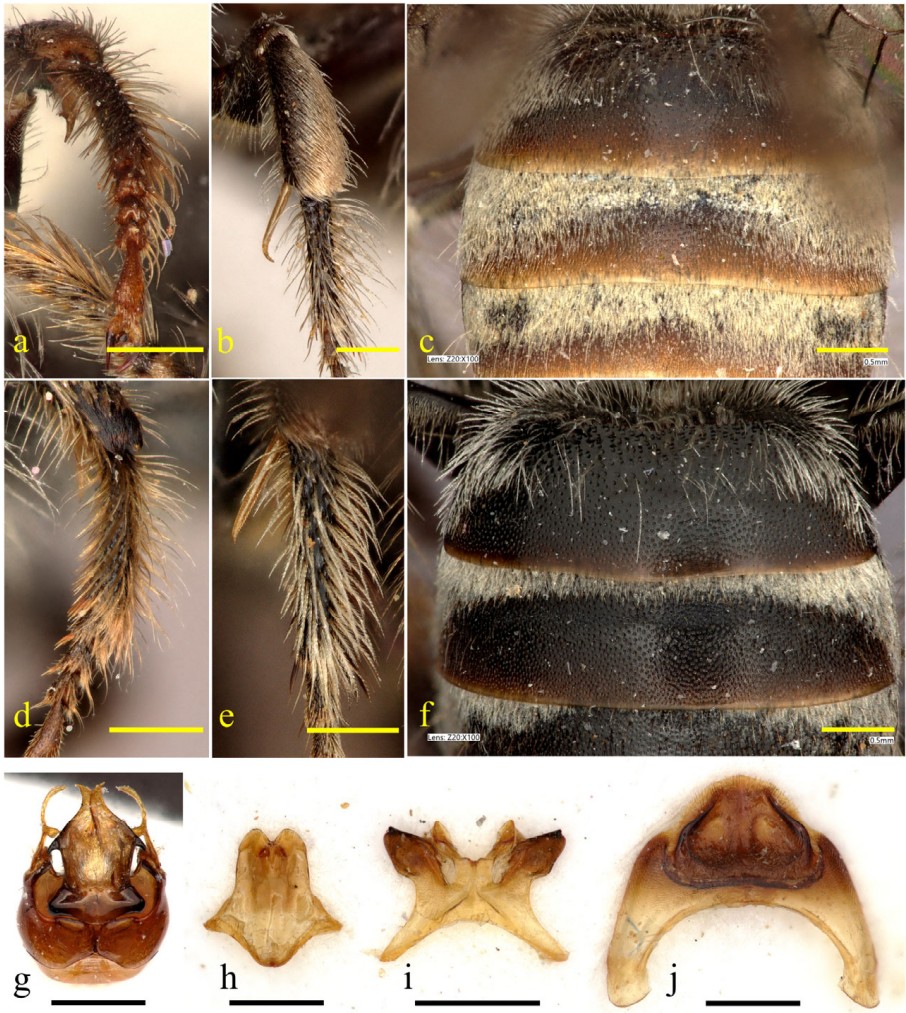

**Figure 8.** Female front tarsi, middle tibial spur, and T1–3 of the *nana*-group. (**a–c**) *Tetralonia epilobii* **sp. nov.** and (**d–f**) *T. nana* Mor., 1874. (**g–j**) Male genitalia and S6–8 of *Tetralonia epilobii* **sp. nov**. Scale bars are 0.5 mm. Collection data of type specimens: (**a–c,g–j**) paratypes, Israel and Palestine, 'En A Tina, 71 m, 33.0783° N/35.6443° E, 29.vii.2011, at *Lythrum salicaria* L., A. Dorchin leg.

In species of the *pollinosa*-group, the scopal hairs are denser, only partly exposing the underlain surface, with most interspaces between the closest hair tubercles being about a tubercle diameter (Figure 7a). This is in contrast to typical *Tetralonia* species that are Asteraceae specialists, in which the scopa is much denser, largely or completely hiding the underlain surface, with the interspaces between the closest hair tubercles being less than a

tubercle diameter (Figure 7f). Also, the hairs on the anterior basitarsus and mediotarsal segments are not particularly specialized in these other species, thinner in *T. malvae*, and shorter in species of the *pollinosa*-group.

**Male:** S6 with posterolateral carinae conspicuous, curved anterolaterally and with anteriorly converging carina, or ridge suggesting *T. cinctella* and *T. eoacinctella* **sp. nov.** (compare Figures 1d and 8j to Figure 3d,h); lateral process of S7 undivided into two lobes, the sclerotized posterior lobe reduced to mere apicolateral point, most reminiscent of *T. scabiosae* (compare Figure 8i to Figure 2c); posteromedial process of S7 short and apically pointed, shortly extended beyond lateral process (Figure 8i); S8 with conspicuous emargination between rounded apical lobes (Figure 8h); gonostylus curvature particularly strong, strongly elbowed, bent ventroapically in lateral view (Figure 1f), slightly impressed on both sides near medially projecting process on inner side, and with apex strongly expanded and almost curved medially, L-shaped as seen in dorsal view (Figure 8g) (the gonostylus is usually angled apicomedially, not curved, in most other species with elbowed gonostylus); maxillary palpus 6-segmented.

### 3.2.2. Taxonomy

#### *Tetralonia nana* **Morawitz, 1874**

*Tetralonia nana* Morawitz, 1874 [27] (p. 144). ♀♂, "Derbent" (Dagestan Republic, Russia). Lectotype: ♂, ZINSP, designated by Proshchalykin et al. [21] (p. 38).
*Macrocera griseola* Pérez, 1879 [26] (p. 150). ♀, "Bordeaux" (France). Lectotype: MNHN, designated by [5] (p. 29). Synonymy in Dalla Torre [22] (p. 241), Friese [23] (p. 85).
*Tetralonia tenella* Mocsàry, 1879 [13] (p. 235). ♂, "In Hungária meridionali-orientali". Type material unlocated, not found in MTM. Synonymy in Dalla Torre [22] (p. 241), Friese [23] (p. 85), based on original syntype collected in "Tasnad" (Tăşnad, Romania).

### 3.2.3. Description of New Species

#### *Tetralonia epilobii* **Dorchin sp. nov.**

**Holotype:** ♂, ISRAEL AND PALESTINE, 'En A Tina, 71 m, 33.0783° N/35.6443° E, 29.vii.2011, at *Lythrum salicaria* L., A. Dorchin leg. (SMNH).

**Paratypes:** ISRAEL AND PALESTINE: 2♂, 'En A Tina, 71 m, 33.0783° N/35.6443° E, 25.x.2010, at *Mentha longifolia* L., A. Dorchin leg. (2♂ADCM); 20♂2♀, 'En A Tina, 71 m, 33.0783° N/35.6443° E, 29.vii.2011, at *Epilobium hirsutum* L., *Lythrum salicaria* L., A. Dorchin leg. (3♂OLML, 3♂SMNH, 14♂2♀ADCM); 6♂2♀, 'En A Tina, 71 m, 33.0783° N/35.6443° E, 22.ix.2012, at *Epilobium hirsutum* L., *Lythrum salicaria* L., A. Dorchin leg. (6♂2♀ADCM); 6♂♀, Nahal Zippori 480 m SW Harduf, 85 m, 32.7622° N/35.1641° E, 7.x.2012, at *Lythrum salicaria* L., A. Dorchin leg. (6♂♀ADCM); 5♂4♀, Nahal Zippori 480 m SW Harduf, 85 m, 32.7622° N/35.1641° E, 12.x.2012, at *Epilobium hirsutum* L., A. Dorchin leg. (♂2♀OLML, ♂♀SMNH, 3♂2♀ADCM); 9♂, 'En A Tina 4 Km S Gonen, 72 m, 33.0844° N/35.6418° E, 27.viii.2016, at *Epilobium hirsutum* L., A. Dorchin leg. (2♂OLML, 2♂SMNH, 5♂ADCM). IRAN: ♀, Fars, Eqlid, 2266 m, 30.8833° N/52.6758° E, 3.viii.2010, R. Khodaparast, leg. (♀ADCM); ♂2♀, Kohgiloyeh & Boyerahmad, Chitab, 1650 m, 30.8016° N/51.2969° E, 22.viii.2019, at *Mentha* sp., T. Hosseini, N. Razmjo, leg. (2♀ADCM); ♀, Yasuj, Ganjee, 1872 m, 30.7619° N/50.3580° E, 16.viii.2016, Z. Alizadeh, leg. (♀ADCM).

**Additional material:** ISRAEL AND PALESTINE: 2♂, 'En A Tina, 71 m, 33.0783° N/35.6443° E, 25.x.2010, at *Mentha longifolia* L., A. Dorchin leg.; 3♂, 'En A Tina, 71 m, 33.0783° N/35.6443° E, 29.vii.2011, at *Epilobium hirsutum* L., *Lythrum salicaria* L., *Mentha longifolia* L., A. Dorchin leg. IRAN: ♂, Kohgiluyeh-Va Boyer-Ahmad, Yasuj, Mazekharide, 1757 m, 30.6386° N/51.5183° E, 16.vi.2017, S.A.A. Hashemi, leg.; ♂, Kohgiluyeh-Va Boyer-Ahmad, Yasuj, Chitab, 1646 m, 6.viii.2019, E. Rostami, leg.; 2♀, Yasuj, Ganjee, 1872 m, 30.7619° N/50.3580° E, 16.viii.2016, Z. Alizadeh, Z. Najafi, leg.; 2♂2♀, Yasuj, Zirtul University, 1796 m, 30.6544° N/51.5872° E, 25,28.vii.2016, F. Malek Hosseini, F.Rajaee, N. Najafi, leg.; ♂, Kohgiloyeh & Boyerahmad, Chitab, 1650 m, 30.8016° N/51.2969° E, 22.viii.2019, at *Mentha* sp., T. Hosseini, leg.; ♀, Sisakht, Cheshmeh mishi, 2484 m, 30.8627° N/51.4872° E, 1.viii.2013,

N. Najafi, leg.; ♂♀, Sisakht, Cheshmeh mishi, 2484 m, 30.8627° N/51.4872° E, 1.viii.2016, Z. Alizadeh, Z. Najafi, leg.; ♀, Sisakht, Cheshmeh mishi, 2484 m, 30.8627° N/51.4872° E, 28.viii.2016, Z. Najafi, leg.; 2♂, Yasuj, Mokhtar, 2150 m, 30.6775° N/51.5183° E, 30.v.2018, Z. Rahimi, leg.

Diagnosis and description:

**Female:** front basitarsus and mediotarsal segments with elongated curved unbranched hairs, on anterior margins with particularly thickened strongly curved long hairs, strongly differentiated from hairs on dorsal surface (Figure 8a); middle tibial spur hooked apically, extended over basal half of basitarsus (Figure 8b); T1 and T2 with impunctate apical margin translucent and broad, on T1 > 8 nearby puncture diameters wide, on T2 4–5 nearby puncture diameters wide (Figure 8c); stipital teeth spacing wide, up to 3–4 basal tooth diameters (Figure 7g); scopal hairs with inconspicuous fine branches (Figure 7b); antenna ferruginous ventrally on flagellar segment 3 onwards (Figure 9b); T2–5 with basal tomentum light fulvous, progressively covering greater portions, comparatively greater in extent, covering almost entirely marginal zones of T3–5 in fresh specimens, with fine unbranched brown hairs on marginal zones (compare Figures 8c and 9a to Figure 8f) and with prepygidial fimbria light brown medially.

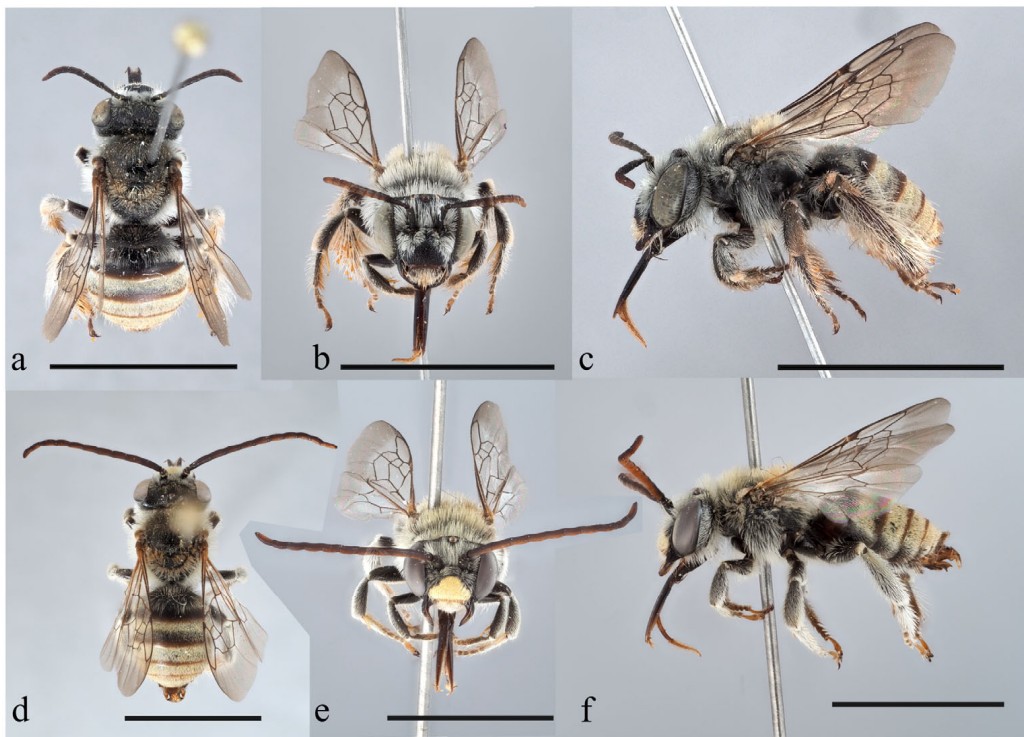

**Figure 9.** Female and male habitus of *Tetralonia epilobii* **sp. nov**. (**a**–**c**) female in dorsal, frontal, and lateral view and (**d**–**f**) male holotype in dorsal, frontal, and lateral view. Scale bars are 5 mm. Collection data of type specimens: Israel and Palestine, (**a**–**c**) paratype, Nahal Zippori 480 m SW Harduf, 85 m, 32.7622° N/35.1641° E, 12.x.2012, at *Epilobium hirsutum* L., A. Dorchin leg.; (**d**–**f**) holotype, 'En A Tina, 71 m, 33.0783° N/35.6443° E, 29.vii.2011, at *Lythrum salicaria* L., A. Dorchin leg.

In females of *Tetralonia nana* Morawitz, the alternative character states are: front tarsi with elongated curved unbranched hairs, with hairs on the anterior margin not strongly differentiated in length or thickness from hairs on the dorsal surface (Figure 8d); middle tibial spur is unmodified, weakly curved apically, hardly attaining the apical half of the basitarsus (Figure 8e); T1 and T2 with an impunctate apical margin opaque, dark, and narrow, on T1 4–5 nearby puncture diameters wide, on T2 2–3 nearby puncture diameters wide (Figure 8f); the stipital teeth spacing is narrower, <2 basal tooth diameters (Figure 7h); scopal hairs with fine conspicuous branches (Figure 7c); antenna dark, dark reddish brown

apically; basal tomentum pale grey largely restricted to the bases of T2 (Figure 8f) and T3, covering almost the entirety of T4, with fine unbranched dark brown hairs on the marginal zones, and with the prepygidial fimbria dark brown medially.

**Male:** clypeus with yellow plus-shape maculation sharply delineated (Figure 9e); labrum completely ivory white (Figure 9e); stipes with some teeth spaces >2 basal tooth diameters; S5 with smooth weakly depressed and impunctate surface apicomedially between dense tufts of thickened sclerotized golden hairs that project posteriorly and hide underlain surface (Figure 1e); middle tibial spur hooked apically as in female (cf. Figure 8b); S6 with posterolateral carinae converged with, and curved anteromedially onto posterior end of anterior carina, both comparatively high and strongly sclerotized, most reminiscent of *T. cinctella* (compare Figure 8j to Figure 2d); lateral process of S7 produced apicolaterally, comparatively slender (Figure 8i); maxillary palpus clearly 6 segmented; T2–5 with extensive light fulvous basal tomentum in fresh specimens, as in females (Figure 8d,f).

In the males of *Tetralonia nana* Morawitz, the alternative character states are: clypeus with yellow, blunt, plus-shape maculation (France and Spain) or reduced, subtriangular maculation (Romania); labrum yellow (France), with wide dark margins (Spain and Romania), or further reduced to a medial inverted triangle (Romania); the stipital teeth spacing is narrow, at most from 1 to <2 basal tooth diameters; S5 with a smooth, weakly depressed, and impunctate surface apicomedially between comparatively sparse tufts of unbranched light to dark brown hairs that project posteriorly and do not completely hide the underlain surface (Figure 1d); the middle tibial spur is unmodified, weakly curved apically, as in female (cf. Figure 8e); S6 with posterolateral carinae that are comparatively low, curved anteromedially onto a blunt anterior ridge (Figure 1d); the lateral process of S7 is comparatively broad; the maxillary palpus is clearly 6-segmented, the two apical segments are small and sometimes weakly differentiated; T2–5 with pale grey basal tomentum comparatively smaller in extent and with fine dark hairs posteriorly, as in female, progressively covering greater portions, with the marginal zones of T2–4 largely exposed in fresh specimens (cf. Figure 8f).

**Etymology:** The new species is named after its known host plant, *Epilobium hirsutum* [9], and is a noun in the genitive form.

**Distribution:** The new species is known so far only from several localities in the North of Israel and Palestine and the Boyer-Ahmad province, around Yasuj, in Iran.

*3.3. Tetralonia Malvae (Rossi, 1790)*

3.3.1. Diagnosis

The species is associated with the Malvaceae [9,31], and females exhibit several adaptations for the collection and manipulation of the large pollen grains of that plant family. They are characterized by sparse scopal hairs, sparse enough to uncover the underlain dorsal surface of both the tibia and basitarsus, arising from tubercles that are spaced apart from the closest tubercles by mostly more than one tubercle diameter (Figure 7d). The maxillary palpus is reduced more so than in the *nana*- or *pollinosa*-groups, 4- (Figure 7i), or 5-segmented with a minute terminal segment, but sometimes with a conspicuous terminal segment, or even approaching 6 segments. The stipital comb spacing is denser than that in the *nana*- and *pollinosa*-groups, with the maximal interspaces between adjacent comb teeth being about a basal tooth width (compare Figure 7i to Figure 7g,h).

In males, S6 has fine oblique posterolateral carina converged onto the anterior end of a blunt anterior carina or ridge, suggesting *T. salicariae* (Lep.) (compare Figure 10d to Figure 10h), *T. glauca* (Fabricius, 1775) [32], and *T. coangustata* Dours. S7 has the lateral process divided by deep emargination into two lobes, and the posterior lobe is sclerotized and slightly elevated, also suggesting the above-mentioned species, but the anterior lobe is rounded basally compared to being bluntly or sharply angled in those species (compare Figure 10c to Figure 10g); S8 has the apical processes bluntly angled on both sides of the posterior emargination, as compared to being rounded in those same species (compare

Figure 10b to Figure 10f). The gonostylus is comparatively weakly curved, arched in lateral view, as compared to elbowed in other species (compare Figure 1g to Figure 1c,f), sinusoidal, without a medial angle, as seen in dorsal view (Figure 10a), suggesting *T. dentata* (Klug, 1835) [18] and *T. graja* (Eversmann, 1852) [20]. The antenna is short, conspicuously shorter than the forewing, 3.3× as long as the compound eye (Figure 11a,c) compared to about as long as the forewing, and 3.8× and 3.6× as long as the compound eye in *T. glauca* and in *T. epilobii* **sp. nov.**, respectively. The first flagellar segment is comparatively short, about as long as that in *T. glauca* and in *T. epilobii* **sp. nov.**, in which it is 2.5× and 2× as long as the second, respectively. The antennae are much shorter in the Afrotropical *macrognatha*-group of species, 2.3–2.7× as long as the compound eye, and the first flagellar segment is longer, 0.6–0.8× as long as the second. This is in keeping with phylogenetic results, which identified *T. malvae* as being more closely related to the remaining Palaearctic species than to this Afrotropical group of species [2].

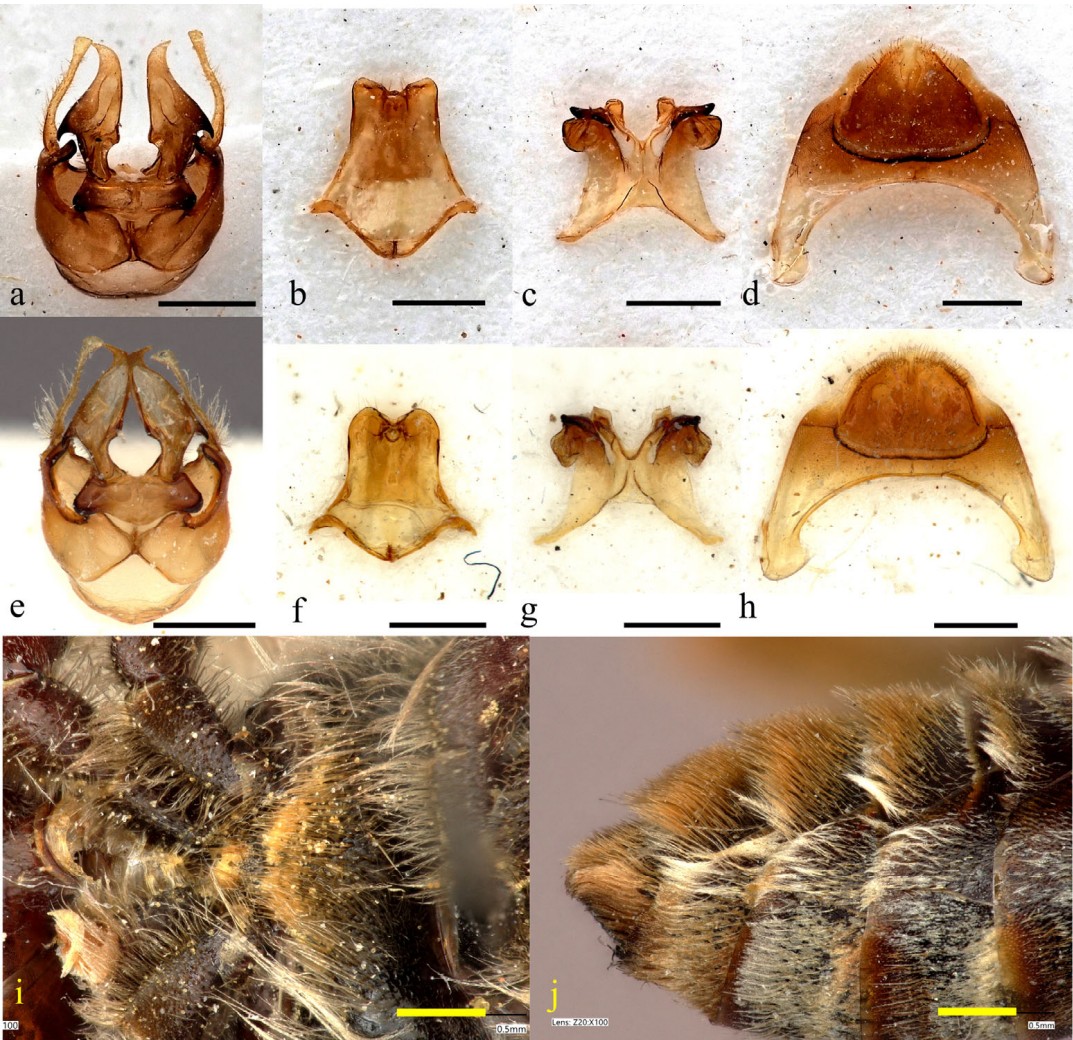

**Figure 10.** Male genitalia and S6–8, and female underside of mesosoma and profile of metasoma. (**a**–**d**) *Tetralonia malvae* (Rossi, 1790) and (**e**–**j**) *T. salicariae* (Lep., 1841). Scale bars are 0.5 mm.

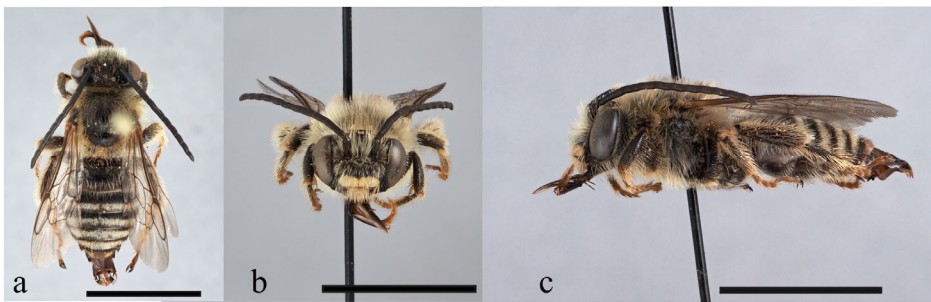

**Figure 11.** Habitus of male neotype of *Apis malvae* Rossi, 1790, in (**a**) dorsal, (**b**) frontal, and (**c**) lateral view. Scale bars are 5 mm. Collection data of neotype: Italy, Lazio, Rome, Maccarese, 30–65 m, 41.8358° N/12.2655° E, 14.vi.2022, M. Selis leg.

### 3.3.2. Taxonomy

*Apis malvae* Rossi, 1790 [10] (p. 107). ♂, "in provinciis Florentina et Pisana" [Florence (Firenze) and Pisa provinces, Italy]. Neotype: Lazio (Roma), Maccarese, OLML, designated here.
*Eucera antennata* Fabricius, 1793 [33] (p. 345). ♂, "in Europa". Neotype: "Lazio, Valle dell'Insupherata (Roma)", NHMD, designated by Michener [34] (p. 19). Type species of *Macrocera* Latreille, 1810 [35] (nec Meigen, 1803 [36]) = *Tetralonia* Spinola, 1838 [1]. Synonymy in Dalla Torre [22] (p. 239), Friese [23] (p. 93).
*Eucera macroglossa* Illiger, 1806 [37] (p. 133). ♂, "Deutschland". Synonymy in Michener [34] (p. 18). ?Syntypes 2♂, MNB [38] (p. 386).
*Macrocera albida* Lepeletier, 1841 [12] (p. 98). ♂, "Espagne". Lectotype: OUM, designated by Baker [19] (p. 1197). Synonymy in Baker [19] (p. 1197).
*Eucera crinita* Klug, 1845 [39] (table 50, fig. 12). ♀, "In Syriam semel lecta" ["Lebanon" in [40] (p. 199)]. Holotype: MNB, examined by B Tkalců [39] (p. 199). Synonymy in Dalla Torre [22] (p. 240, as *Eucera malvae* var. *crinita* Klug, 1845), Friese [23] (p. 93), Baker [41] (p. 544).
*Tetralonia macroglossa* ssp. *xanthopyga* Alfken, 1936 [42] (p. 106), **Syn. nov**. ♀♂, "Bologna" "Bologna (Ronzano)" (church of San Vincenzo di Ronzano), "Ungarn, Simontornya" (Hungary). Holotype: ♀, "Ungarn, Simontornya" (Hungary), MNB.

### *Apis malvae* **Rossi, 1790**

This is the oldest name which has been traditionally associated with the species, and is confidently traced back at least to Lepeletier [12] (p. 96), [5] (p. 28). It was, however, considered by Bischoff & Hedicke [38] (p. 385) as belonging to a different species (see below). Recently, Scheuchl et al. [43] (p. 50) proposed that this species was likely *Tetralonia nana* Morawitz, arguing that, like *T. malvae*, it is associated with the Malvaceae, and fits the original description better. However, despite the potentially inaccurate measures originally given by Rossi [10] (p. 107), other characteristics that he provided, such as the metasomal hair pattern, are diagnostic of *T. malvae*, and there is importance in preserving this name in its traditional sense [see [34] (p. 18)]. Therefore, under the principle of name stability and for avoiding future synonymy of long used names, in accordance with article 75 of the International Code of Zoological nomenclature [44], a neotype for *Apis malvae* is here designated to fix its traditional species concept. The neotype is a male from the region of Lazio (Rome, Maccarese, 41.8358° N/12.2655° E) in central Italy that was collected on the 14.vi.2022 by M Selis. The neotype is relatively small, about 9 mm long, in keeping with the original description, 'almost half the size of *A. longicornis*' ("Fere dimidio minor *A. Longicorni*") (males of *Apis longicornis* Linnaeus, 1758 [45] = *Eucera longicornis* (Linnaeus) are 12–15 mm long). Also in keeping with the original description, the mesosoma has greyish, fulvous, semierect, finely branched hairs (to light brown in some lights, but without dark hairs), and the metasomal tergites 2–5 have basal hairs of the same color (largely covered by, but seen through the hyaline apex of the preceding tergites), as well as light posterior hair bands that are divided by dark short hairs (Figure 11a,c). In addition, the antennae are

uniformly dark, the clypeus has a yellow irregular apical band, and the labrum is largely yellow (Figure 11b,c).

*Eucera macroglossa* **Illiger, 1806**
Bischoff & Hedicke [38] (p. 386) reported on two male syntypes preserved in MNB. They argued that, since Illiger worked on a revision for Rossi's [10] Fauna Etrusca, he must have known the identity of *Apis malvae* and therefore would not propose another name for the same species. They further claimed that, based on the original description, *Apis malvae* must have been a different species, and therefore the Illiger [37] name should become valid. This view was followed in a recent paper by Scheuchl et al. [43], and we here propose a solution to retain the name '*malvae*' for this species (see above). In any case, '*macroglossa*' remains invalid, since the next available name for this species is *Eucera antennata* Fabricius, 1793 [33] (incidentally, the type species of the genus *Tetralonia*), as shown in the list above. No potential syntypes marked by Illiger or attributed to him have been located so far in MNB (S Krause, pers. com.).

*Tetralonia macroglossa* **ssp.** *xanthopyga* **Alfken, 1936**
Based on its description, this taxon differs from the nominal subspecies merely in hair color. The syntypes preserved in MNB were examined with the help of the collection manager S Krause and they include an additional conspecific female and four males, all from the series collected by Pillich in Simontornya, Hungary. The holotype is marked with a red label printed with "Typus", and both the identification label and citation from the locality label "5.7.31. Pf. e. Auf Torilis Anthriscus." are accurate (except for the latter's last word). The remaining type specimens that are preserved in MNB comprise five specimens and correspond with the male "Allotype" that was collected on "8.8.31, auf Salvia nemorosa", a second female "Paratypus" collected on "11.8.33", and three additional males "Paratypus" collected on "6.8.33", "6.8.33... auf Lavatera thuringiaca", and "4.8.33... auf Lavatera thuringiaca".

*3.4. Tetralonia salicariae (Lepeletier, 1841)*
3.4.1. Diagnosis

This species is associated with pollen of *Lythrum* (L.) [31] and exhibits unique characteristics in the female as well as male, including potential adaptations for the collection of the comparatively small pollen grains of its host plant.

**Female:** The medial underside of the mesosoma and the front and middle coxa and trochanter have unbranched apically undulating golden hairs (Figure 10i); S2–5 have dense stiff unbranched apically curved golden hairs (Figure 10j) (the hairs are finer, shorter basally, and straight apically on the sternites in most other species); the stipital comb is dense, with tooth interspaces of less than a basal tooth width (compare Figure 7j to Figure 7g–i); the prementum has apically curved hairs on the ventral surface directed posteriorly (Figure 7j); and the scopal hairs are unbranched, except for strongly branched hairs along the posterior margins of both the tibia and basitarsus (Figure 7e).
**Male:** Clypeus with yellow blunt plus shape maculation; labrum immaculate, dark, at most with faint medial yellow maculation; and the prementum has apically curved hairs directed posteriorly on the ventral surface, as in the female (cf. Figure 7j). Many structures of the genitalia and associated sternites are reminiscent of *T. malvae* and Asteraceae specialist species, as follows. S6 has fine oblique posterolateral carina converged onto the anterior end of a blunt anterior carina or ridge, suggesting *T. glauca* (F.) and *T. malvae* (Rossi) (compare Figure 10h to Figure 10d), but sometimes both the converging carinae are almost absent; the lateral process of S7 is divided into two lobes by wide emargination, the anterior lobe is angular, and the posterior lobe is strongly elevated and apically sharp (Figure 10g), most reminiscent of *T. glauca* (F.); S8 is deeply emarginated between rounded apical lobes (Figure 10f), suggesting *T. glauca* (F.) and *T. vicina* Morawitz; the gonostylus curvature is comparatively weak, weakly elbowed, as seen in lateral view, without a conspicuous medial angle on the inner side, and is apically spatulate or weakly expanded medially, as

seen in dorsal view (Figure 10e); the mesonotum and tergites have sparse, weakly defined shallow punctures suggesting *T. malvae*; the maxillary palpus is 6-segmented (but a male examined from Macedonia, Greece, has a 5-segmented palpus with a long 3rd segment); and the stipital teeth spacing is narrow with a <1 basal tooth diameter, as in the female (cf. Figure 7j).

### 3.4.2. Taxonomy

*Macrocera salicariae* Lepeletier, 1841 [12] (p. 102). ♀♂, "Environs de Paris; à St.-Séver. Envoyée par M. Léon Dufour sous le nom que je lui conserve." (around Paris; Saint-Sever, south France). Lectotype: ♀, MNHN, designated in Dorchin [5] (p. 30).
*Macrocera meridiana* Lepeletier; Dufour [46] (p. 420) [originally given as "Macrocera meridiana. Lep. (inéd.), ex ipso."]. ♀♂, (no locality given). Synonymy in Pérez [26] (p. 146).
*Tetralonia lythri* Schenck, 1867–1868 [47] (p. 280). ♀♂, "Danzig". New interpretation for *Tetralonia salicariae* (Lepeletier, 1841) in Brischske [48] (p. 3). Synonymy in Pérez [26] (p. 146).
*Tetralonia basalis* Morawitz, 1871 [49] (p. 313). ♂, "Bei Kasan" (Kazan, Russia). Synonymy in Pérez [26] (p. 146). Lectotype: ♂, "Kasan" (Kazan, Republic of Tatarstan, Russia), ZINSP, designated by Proshchalykin et al. [21] (p. 37). Synonymy in Levchenko et al. [50] (p. 322).

## 4. Discussion

The taxa included in this paper comprise an assemblage of species that do not form a single clade, but rather share an atypical, non-Asteracious floral association within the genus *Tetralonia* (figure 2 in ref. [2], figure 1 in ref. [9]). Most of the species included in this work belong to a well-delimitated group, namely the *pollinosa*-group; they share pollen host plants in the family Caprifoliaceae [9] and can be confidently diagnosed in both the female and the male sexes. Two additional species are highly morphologically reminiscent and are placed here in the *nana*-group, although, in this case, they do not share the same pollen hosts (see in the diagnosis of that group). The relationships of the remaining species are less well-defined, and different species exhibit morphological characteristics that are reminiscent of supposedly unrelated non-Asteraceae, as well as Asteraceae pollen specialists. Such reminiscent structures are often observed in the genital complex of the males, and they may have been attained convergently, or perhaps they are plesiomorphic. For example, the species *T. nana* and *T. epilobii* share an elbowed male gonostylus, conspicuous posterolateral carina of S6, and unemarginated lateral process of S7 with *T. scabiosae* and other *pollinosa*-group species, as well as with some Asteraceae specialist species like *T. lyncea* Mocsàry, 1879 [13] and *T. nigrifacies* Dours, 1873 [24] (AD, pers. obs.). *Tetralonia vicina* exhibits the same characteristics, and, in fact, was treated previously as being conspecific with *T. scabiosae* [16], although it has an emarginated process of S7 (see more details in the diagnoses of the *pollinosa*- and *nana*-groups). Additional Asteraceae specialists such as *T. glauca* and *T. coangustata* suggest somewhat intermediate morphologies, including an elbowed gonostylus together with a lower posterolateral carina of S6 and widely emarginated lateral process of S7, the latter of which are reminiscent of two other species studied in this work, *T. malvae* and *T. salicariae* (AD, pers. obs. and see in the accounts of these species). Additional characteristics that are shared among the non-Asteraceae specialists are apparently related to their association with the relatively large pollen grains of their host plants, mainly in the botanical families Caprifoliaceae, Malvaceae, and Onagraceae [9] (*T. salicariae* is a specialist of Lythraceae and is excepted [31]). For example, the particularly sparse scopal hairs of the female and the widely spaced stipital comb are obvious adaptations for trapping larger pollen grains than those typically found in the Asteraceae. Such functional traits are likely to be convergently attained and thus should be considered with caution when inferring systematic relationships [2]. A further taxonomic and phylogenetic study of the genus *Tetralonia* is yet to be performed and would be much needed to delimitate and resolve some systematically difficult groups and poorly known species.

**Author Contributions:** Conceptualization, A.D.; methodology, A.D.; validation, A.D. and D.M.; formal analysis, A.D.; investigation, A.D.; resources, A.D. and D.M.; data curation, A.D.; writing—original draft preparation, A.D.; writing—review and editing, A.D. and D.M.; visualization, A.D.; supervision, A.D. and D.M.; project administration, A.D.; funding acquisition, A.D. and D.M. All authors have read and agreed to the published version of the manuscript.

**Funding:** This research was funded by DG ENV project ORBIT (Taxonomic resources for European bees, contract No. 09.029901/2021/848268/SER/ENV.D.2); the DG ENV project SPRING (Strengthening Pollinator Recovery through Indicators and monitoring, contract No. 09.02001/2021/847887/SER/ ENV.D.2); and FEDtWIN project (contract No. Prf-2019R-042_HYMPOL, BELSPO).

**Data Availability Statement:** The data associated with the species newly described in this paper were deposited in Zoobank, https://zoobank.org. (accessed on 11 January 2024). Present article: urn:lsid:zoobank.org:pub:0670F038-B376-4F11-BC21-BC27A0967989. *Tetralonia eoacinctella* **Dorchin, 2024:** urn:lsid:zoobank.org:act:B6F44D70-8285-4F66-A74E-0DEF9498E229. *Tetralonia epilobii* **Dorchin, 2024:** urn:lsid:zoobank.org:act:DC0EEBE8-A59F-4DB0-A0E1-9C8B63989FCA. *Tetralonia stellipilis* **Dorchin, 2024:** urn:lsid:zoobank.org:act:CE87467C-B528-4501-ABF7-4FA54C1E69B0.

**Acknowledgments:** We thank the collection manager Stefanie Krause (MNB, Germany), and the curator Joseph Monks (BMNH, UK) for providing information on type material that is preserved in their institutes. We are grateful for Alireza Monfared and his research group (Dept. of Plant protection, Faculty of Agriculture, Yasouj University, Iran) for allowing us to examine specimens preserved in their institute. Marco Selis (Viterbo, Italy) has kindly provided the neotype of *T. malvae*, and Paolo Rosa (UMons, Belgium) clarified nomenclatorial considerations related to its designation.

**Conflicts of Interest:** The authors declare no conflicts of interest.

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
