# Peer review of "Species of the Western Palaearctic Genus Tetralonia Spinola, 1838 (Hymenoptera, Apidae) with Atypical Pollen Hosts, with a Key to the pollinosa-Group, Description of New Species, and Neotype Designation for Apis malvae Rossi, 1790†"

_2673-6500, doi:10.3390/taxonomy4010007_

Round 1

Reviewer 1 Report

Comments and Suggestions for Authors

- In general terms the manuscript requires changes.

- Good command of the English language, the manuscript only requires some minor corrections marked in the text.

- All the authors and years of the scientific names are misspelled, they are all missing a comma, even in the title.

- The results are presented in an orderly and systematic way.

- Phylogenetic analyzes are not presented and it would be very convenient to perform them, since we talk about nomenclatural changes, lineages, restitutions, homoplastic characters, synapomorphic characters and without a formal phylogenetic analysis, all groupings will continue to be subjective and artificial.

- Statistical analyzes are not presented because they are not necessary.

- The discussion and conclusions are subjective if a phylogenetic analysis.

- Does not show ascending numerical order in in-text citations.

- Bibliographic citations number 20, 21, 22 and 38 are out of order.

- The citations of reference numbers: 7, 32 and 40 are missing from the text.

- The figures are relevant, but require improvements, such as adding some scale lines, homogenizing the place where the letter of each figure is placed and, above all, writing the figure captions mentioning each image in alphabetical order.

Comments on the Quality of English Language

- Good command of the English language, the manuscript only requires some minor corrections marked in the text.

Reviewer 2 Report

Comments and Suggestions for Authors

A contribution very needed and that must be published.

Round 2

Reviewer 1 Report

Comments and Suggestions for Authors

After corrections the manuscript reads very well.

Author Response

We thank the reviewer for his efforts that helped us improve the manuscript!